# Qualitative and Quantitative Evaluation of Rosavin, Salidroside, and p-Tyrosol in Artic Root Products via TLC-Screening, HPLC-DAD, and NMR Spectroscopy

**DOI:** 10.3390/molecules27238299

**Published:** 2022-11-28

**Authors:** Hanna Nikolaichuk, Marek Studziński, Marek Stankevič, Irena M. Choma

**Affiliations:** 1Department of Chromatography, Faculty of Chemistry, Maria Curie-Sklodowska University, Maria Curie-Sklodowska Sq.3, 20031 Lublin, Poland; 2Department of Bioanalytics, Faculty of Biomedicine, Medical University of Lublin, Jaczewskiego St. 8b, 20090 Lublin, Poland; 3Department of Physical Chemistry, Faculty of Chemistry, Maria Curie-Sklodowska University, Maria Curie-Sklodowska Sq.3, 20031 Lublin, Poland; 4Department of Organic Chemistry, Institute of Chemical Sciences, Faculty of Chemistry, Maria Curie-Sklodowska University, Gliniana St. 33, 20613 Lublin, Poland

**Keywords:** TLC-screening, HPLC-DAD, NMR, arctic root

## Abstract

Artic root is a well-known plant adaptogen with multipotential pharmacological properties. Thin-layer chromatography (TLC)—screening followed by diode-array high-performance liquid chromatography and nuclear magnetic resonance spectroscopy proved to be a reliable and convenient method for the simultaneous determination of the quality of various herbal raw materials and supplements. This combination allowed for comparing and differentiating arctic root samples as well as defining their authenticity. The study provided information on the chemical and biological properties of the seven chosen samples as well as qualitative and quantitative evaluation of the quality markers: rosavin, salidroside, and p-tyrosol. The absence of rosavin, salidroside, and p-tyrosol in three samples was detected using TLC screening and confirmed by HPLC-DAD and NMR. The paper highlighted the importance of quality control and strict regulation for herbal medicine supplements and preparations.

## 1. Introduction

Artic root, also known as *Rhodiola rosea* (*R. rosea*), is a worldwide known medicinal and functional plant with multipotential healing abilities. It has been used in European and Asiatic traditional medicines primarily as an adaptogen, which means that arctic root helps a human organism to produce a non-toxic and non-specific response to physical, biological, and chemical stressors [1,2]. This activity does not disturb homeostasis and has normalizing and health-promoting effects regardless of changes caused by the stressors. With the development of pharmaceutical technology, the application of *R. rosea* rhizome and root has been expanded. Arctic root material is added to various products, including food supplements and health and beauty products, which are increasingly accepted in European countries. In recent years, the research on arctic root has mainly concerned chemical and biological profiling as well as pharmacological activity. Various biological activities of arctic root, including antioxidant, antibacterial, antiviral, anticancer, immunostimulant, antidepressant, and ergogenic, were identified [3,4,5,6]. In clinical practice, arctic root has been investigated to improve physical and mental performance and treat fatigue and depression.

The safety and quality of herbal medicines available to European consumers have been vital for medicine regulations. The introduction of the Traditional Herbal Medicinal Products Directive (formally Directive 2004/24/EC) and herbal registration gave confidence to consumers that they have access to a wide range of popular herbal medicines with high quality and safety. However, many unregulated drugs are still widely available on the market [7]. Primarily, this concerns economically profitable products when raw plant material shows high bioactivity, short expiry dates, and possible enzymatic degradation. Arctic root is one of these kinds of items. Moreover, due to the popularity of arctic root products among sportspersons, they may be adulterated with performance-enhancing synthetic stimulants, e.g., steroids. It should also be mentioned that other cheap species of *Rhodiola,* such as *R. crenulata*, *R. kirilowii*, *R. quadrifida*, *R. sacra*, and *R. yunnanensis*. *R. crenulata* are widespread in China and thus can be of main economic interest. *R. crenulata* is cheaper than arctic root and relatively easy to gather, which causes the falsification of *R. rosea* products. Therefore, the potential risks of this kind of adulteration are lack of effectiveness and beneficial effect on organisms [8,9].

This study used TLC—screening, HPLC-DAD, and ^¹^H-NMR spectroscopy to compare the identity and composition of the arctic root products. These techniques provide complementary data, which allowed us to discriminate among tested samples. TLC analysis gives qualitative data to determine the marker compounds and made visual comparisons among different products relatively quickly. Moreover, TLC-bioassays enable an estimation of the biological properties of the samples.

The focus was on the presence of rosavin, salidroside, and p-tyrosol in the tested arctic root samples using the methods mentioned above. Since these constituents play a critical role in arctic root bioactivity and are regarded to be responsible for their pharmacological properties, they are used to standardize the raw materials. The lack of marker compounds indicates possible adulteration and poor quality of samples. HPLC and spectroscopic methods confirmed the obtained TLC results.

## 2. Results and Discussion

### 2.1. TLC Screening

TLC screening, a fast and straightforward tool, was used to compare visual differences between the chosen arctic root samples, which may be caused by adulteration or poor quality. The content of rosavin, salidroside, and p-tyrosol in arctic root depends on various factors such as a different origin, harvest, age of the plant, method of raw material stabilization, storage conditions, and even the extraction method.

Available literature data on the last item, i.e., solvent and extraction method, are contradictable and mainly concern the aqueous and ethanol extraction [10]. However, our results indicated that 70 % methanol extract from a maceration of the dry arctic root shows the highest bioactivity among water, ethanol, methanol, and ethyl acetate [11,12]. Additionally, in our previous work [11], TLC screening pointed to the absence of rosavin, one of the major compounds and quality markers, in one of the tested samples. To acquire more information, the AS, thymol, NP-PEG, DPPH (chemical screening), and tyrosinase, lipase, and α-glucosidase (biochemical screening) assays were done for seven arctic root samples (including United States Pharmacopeia standard of *R. rosea* root and rhizome denoted by S6) together with the standards: rosavin—authenticity marker, and salidroside and p-tyrosol as bioactivity markers. The obtained results (Figure 1 and Figure 2) pointed to the high variability of constituents (sugars, glucosides, polyphenols, etc.—see AS, thymol, and NP-PEG), different inhibition activities against enzymes (lipase, tyrosinase, and α-glucosidase assays) and different antioxidant (DPPH assay) properties. The S2, S4, and S5 samples showed potent inhibition of lipase, α-glucosidase, and tyrosinase, as well as strong antioxidant activities, whereas the S1, S6, and S7 samples revealed weaker inhibition of lipase, tyrosinase, and α-glucosidase along with the antioxidant activity. The S3 sample differs in chemical and biological profiles from the other samples, which could be explained by the low content of *R. rosea* material or the falsification of this supplement.

The band of the rosavin standard was visible at *hR_F_* 24 under UV 254 nm and after chemical visualization with AS and thymol reagents (Figure 1). Moreover, rosavin revealed inhibition of tyrosinase and α-glucosidase at 5 µg (Figure 2). Based on these chromatograms, it can be concluded that rosavin was present only in three samples (S1, S5, S6). The absence of rosavin in the S2 was already confirmed in our previous paper by HPLC-ESI-MS, which suggested the adulteration of that product or improper manufacturing procedures [11]. However, in the S3, S4, and S7 samples, the absence of rosavin had to be confirmed (it is possible that the amount of rosavin in samples could be under the detection limit in the assays).

Similarly to rosavin, the presence of salidroside and p-tyrosol compounds are tough to evaluate in arctic root samples using only TLC screening. The salidroside standard band at *hR_F_* 47 was visible at 254 nm as well as in AS and thymol assays. Moreover, the 5 µg amount of salidroside showed antioxidant activity as well as tyrosinase and α-glucosidase inhibition. Salidroside was visible in four samples (S1, S4, S5, and S6) in AS assay under 366 nm. Concerning p-tyrosol, related bands were visible at *hR_F_* 78 at 254 nm and in the AS assay. The marker showed antioxidant and enzyme inhibition (tyrosinase and α-glucosidase) activities at 5 µg. The presence of p-tyrosol was stated in four samples, S1, S2, S5, and S6, in AS assay at 366 nm. Rosavin, salidroside, and p-tyrosol were not detected using the NP-PEG test. Besides, none of the markers showed lipase inhibition activity at 5 µg. In order to verify TLC screening results and observations regarding the presence of marker constituents in arctic root samples, HPLC-DAD and NMR were used.

### 2.2. HPLC-DAD Analysis

The HPLC-DAD method was used to detect and quantify four marker components (rosavin, salidroside, and p-tyrosol) in seven arctic root samples. The results indicated differences in the presence of markers and their amounts in the *R. rosea* samples. The contents of the markers in the extracts were calculated from the calibration curves constructed for four standards. The calibration ranges, regression equations, correlation coefficients, and LOD and LOQ values are given in Table 1.

HPLC-DAD analysis confirmed the results from TLC screening regarding the presence/absence of rosavin, salidroside, and p-tyrosol in arctic root extracts (Table 2). The TLC results concerning the absence of rosavin in S2, S3, S4, and S7 samples were confirmed. Only three samples (S1, S5, and S6) contain all three markers, while the S3, S4, and S7 samples have none. The content of rosavin, the authenticity marker of arctic root extracts, varied between 59.99 ± 3.74 µg/mL (S1) and 100.46 ± 5.44 µg/mL (S5). The amount of salidroside was the highest in the reference standard S5 (37.86 ± 3.94 µg/mL), followed by S1 and S6 samples, while the lowest content was in the S2 sample. The quantities of p-tyrosol varied between 2.61 ± 0.52 µg/mL (S6) and 4.94 ± 0.12 µg/mL (S5). That differs from the results of Marchev et al. [9], where only trace amounts of p-tyrosol were detected in *R. rosea* samples. Salidroside and p-tyrosol were not detected in S3, S4, and S7. The HPLC-DAD results suggest that the contents of the marker constituents of *R. rosea* depend on the morphology of plant material (rhizome or/and root) and the type of products (supplements, preparations).

### 2.3. NMR

For the more reliable identification of the compounds, NMR analysis was used. First, ^1^H NMR spectra of standard compounds: rosavin, salidroside, and p-tyrosol (Figure 3) were obtained (Figure 4). This allowed identifying the most characteristic signals for each compound for the subsequent analysis of extracts. The seven samples of arctic root were analyzed by ^1^H NMR, and the fingerprint profiles were obtained (Figure 5 and Figure 6). The results revealed qualitative and quantitative differences among arctic root samples. Based on the characteristic signals of marker compounds in specific regions of NMR spectra, rosavin, salidroside, and p-tyrosol were identified (detailed information in Table 3).

As seen from ^1^H NMR spectra of analyzed extracts and standards, three samples (S1, S5, and S6) undoubtedly contain rosavin, salidroside, and p-tyrosol). In three other samples, S3, S4, and S7, the amounts of the markers are very low or non-detectable. Sample S2 lacks rosavin, as was also proved by TLC, HPLC-DAD, and previous HPLC-ESI-MS results [11]. The detection of p-tyrosol in the analyzed samples was also challenging as the signals in aromatic regions overlay with those from the salidroside. In general, small amounts of the investigated compounds are difficult to confirm due to the lower sensitivity of NMR compared to chromatographic methods. Additionally, the presence of a complex matrix may strongly influence the chemical shifts of hydrogen atoms present in the molecules; therefore, a simple comparison with a pure standard rarely gives a sure answer.

## 3. Materials and Methods

### 3.1. Reagents and Materials

Acetone (99.5%), acetic acid (99.5–99.9%), diphenylboryloxyehtylamine (NP) (≥97.0%), ethanol (96%), ethyl acetate (99.8%), 2-isopropyl-5-methyl-phenol (thymol) (≥98.5%), methanol (99.8%), o-phosphoric acid (85%), polyethylene glycol—4000 (PEG-4000), phosphate buffer, sodium hydroxide (≥98%), sulfuric acid (96–98%), and sodium acetate buffer were from POCH (Gliwice, Poland). P-Anisaldehyde (AS) (≥98%), bovine serum albumin (BSA) (≤100%), 2,2-diphenyl-1-picrylhydrazyl (DPPH) (≤100%), 3,4-dihydroxy-L-phenylalanine (L-DOPA) (>98.0%), α-glucosidase from *Saccharomyces cerevisiae* (≥10 units/mg protein)*,* lipase from porcine pancreas (30–90 units/mg protein), 1-naphthyl acetate (≥98%), 2-naphthyl acetate (≤100%), 2-naphthyl α- D glucopyranoside ((≤100%), polyethylene glycol tert-octylphenyl ether (Triton X) (≤100%), rosavin (≥98.0%), salidroside (≥98.0%), p-tyrosol (98%), Fast Blue B salt (95%), tris(hydroxymethyl)aminomenthane hydrochloride (TRIS) buffer, and tyrosinase from mushroom (≥1000 unit/mg solid) were purchased from Sigma Aldrich (Poznań, Poland). Pure water was from the Millipore Q system (Millipore, Bedford, MA, USA). All reagents were of the analytical grade. Acetonitrile used in HPLC-UV/Vis-DAD quantification procedure was purchased from Romil (Cambridge, UK). TLC silica gel 60 F_254_ (20 × 10 cm) plates were purchased from Merck, (Darmstadt, Germany).

### 3.2. Sample Preparation

The seven samples (Appendix A) of arctic root were purchased from different Polish vendors; one of the samples is the United States Pharmacopeia reference standard of root and rhizome from Sigma Aldrich (Poznań, Poland) (S5). The samples (0.1 g) were filled with 70 percent methanol (1 mL), and maceration took place in the darkness at room temperature for 72 h. After maceration, the extracts were filtered through a paper filter. Obtained extracts were used for TLC application. The standards of rosavin, salidroside, and p-tyrosol were prepared at a concentration of 1 mg/mL in methanol for TLC analysis. All samples were stored at −8 °C.

### 3.3. TLC-Chromatography

The TLC chromatograms were obtained as follows. Samples application was conducted using the automatic TLC applicator Linomat 5 (Camag, Muttenz, Switzerland). On the NP-TLC plate, 5 μL of samples (plant extracts and standards) were applied as 8 mm bands (10 mm from the lower and left edge, at a distance of 13 mm between tracks). As the mobile phase, ethyl acetate-methanol-water 77:13:10 (*V*/*V*/*V*) was used. The development of the mobile phase was up to 8 cm in the DS sandwich chamber (Chromdes, Lublin, Poland). After development, chromatograms were dried on a cold stream of air for 20 min. Documentation of chromatograms was at UV 254 nm, UV 366 nm, and white light illumination (at the reflectance mode) using Visualiser with DigiStore 2 Documentation System, VideoScan 1.1, and winCATS 1.4.7 software (Camag, Muttenz, Switzerland). All chromatograms were prepared in the same way under the same conditions and further used for chemical and biochemical assays. The assay reagents/solutions were sprayed on TLC plates using automatic piezoelectric spraying (TLC Derivatizer for 20 × 20 cm plates, Camag, Muttenz, Switzerland).

### 3.4. TLC-Chemical Screening

#### 3.4.1. AS Reagent Assay

AS reagent was prepared as follows 0.1 mL of p-anisaldehyde was dissolved in 17 mL of methanol, then to the mixture were added 2 mL of acetic acid and 1 mL of sulfuric acid. The plate was sprayed with 4 mL (red nozzle, speed 6) of AS reagent and then heated at 105 °C for 7 min on the TLC Heater (Camag, Muttenz, Switzerland). The results were documented at VIS and UV 366 nm light. AS, an excellent general reagent, is used to detect organic compounds such as terpenes, terpenoids, saponins, sugars, and propylpropanoids [13].

#### 3.4.2. Thymol Reagent Assay

Thymol reagent solution was prepared by dissolving 0.1 g of thymol in 19 mL of ethanol and mixing it with 1 mL of sulfuric acid. The plate was sprayed with 4 mL (red nozzle, speed 6) of thymol reagent, and the plate was heated at 120 °C for 15 min. The results were documented at VIS light [13].

#### 3.4.3. NP-PEG Reagent Assay

The plate was sprayed (green nozzle, speed 6) with 4 mL of 10 mg/mL NP methanol solution. After drying, the plate was sprayed (green nozzle, speed 6) with 4 mL of 50 mg/mL PEG ethanol solution. The results were documented at UV 366 nm light. NP-PEG reagent is used for the detection of polyphenols [13].

#### 3.4.4. DPPH Assay

The plate was sprayed (blue nozzle, speed 6) with 4 mL of 0.2% DPPH methanol solution. Results were documented after 30 min at the VIS light. Radical scavengers appeared as white bands against the purple background [14].

### 3.5. TLC-Biochemical Screening

#### 3.5.1. Tyrosinase Bioassay

The plate was sprayed (red nozzle, speed 6) with 2.5 mL of substrate solution (0.1183 g of L-DOPA diluted in 49.5 mL of 0.02 M phosphate buffer (pH 6.8) and added 0.5 mL of Triton X). Subsequently, the plate was sprayed (red nozzle, speed 6) with 3.0 mL of enzyme solution (400 units of tyrosinase in 1 mL of 0.02 M phosphate buffer, pH 6.8). The incubation lasted 10 min at room temperature in the closed vessel with a humid atmosphere in a dark place. Results were documented at VIS light. Tyrosinase inhibitors are revealed as white bands on grey background [15].

#### 3.5.2. Lipase Bioassay

The plate was sprayed (green nozzle, 6 speed) with 3 mL of 1.5 mg/mL of 1-naphthyl acetate of ethanol solution. After that, the plate was sprayed (red nozzle, speed 6) with 4 mL of enzyme solution (50 units of lipase and 5 mg of BSA dissolved in 5 mL of 0.05 M TRIS buffer, pH 7.4). The incubation was at 37 °C for 20 min in a closed vessel in a humid atmosphere. After incubation, the plate was sprayed (blue nozzle, 6 speed) with 2 mL of 0.5 mg/mL of Fast Blue B salt water solution. Results were documented at VIS light. Lipase inhibitors emerged as white bands on a purple background [16].

#### 3.5.3. α-Glucosidase Bioassay

The plate was sprayed (green nozzle, speed 6) with 2 mL of 1.2 mg/mL 2-naphthyl α- D glucopyranoside ethanol solution. Then, the plate was sprayed (red nozzle, 6 speed) with 4 mL of enzyme solution (50 units of α-glucosidase dissolved in 5 mL of sodium acetate buffer, pH 7.5). The plate was incubated in the humid atmosphere at 37 °C for 10 min in a closed vessel. After incubation, the plate was sprayed (blue nozzle, speed 6) with 0.5 mL of 1 mg/mL of Fast Blue B salt aqueous solution. Results were documented at VIS light. α-Glucosidase inhibitors are visible as white bands on the purple background [17].

### 3.6. HPLC-DAD

The HPLC-UV/VIS-DAD chromatograms were collected using the following equipment purchased from Shimadzu Corporation (Kyoto, Japan): control unit CBM-20A, two pumps LC-20AD (high-pressure gradient), degasser DGU-20A5R, autosampler SIL-20AC HT, column oven CTO-20AC, and detector SPD-M20A. Column: Phenomenex Kinetex C18 100, dimensions: 4.6 mm × 150 mm. The system was working under the control of LabSolutions software (Shimadzu Corporation) Version 5.71SP2. The mobile phase was water (A) and acetonitrile (B) with a flow rate—of 1 mL/min. The injection volume was set to 10 μL for the standards and 1 μL for the extracts. Each run started at 10% of B and increased to 12% in 7 min and then 100% in 25 min. The column temperature was set at 30 °C. In order to purge the highly lipophilic compounds present in the extracts, the flow was held from 25 min to 45 min at 100% B, at 60 °C. Then, the B concentration was linearly lowered to 10%, and the temperature was cooled down to 30 °C to restore initial conditions. Every development was triplicated, and the presented results are averages from the obtained results (Appendix A). The RSD values of retention times and peak areas were lower or equal to 5%. Diode-array detection was set to collect data at 218 nm and 254 nm.

### 3.7. NMR

The ^1^H NMR spectra (zg30 pulse program) were recorded with Bruker Ascend 500 MHz spectrometer in MeOH-d4 (Deutero, 99.8 atom %D) as a solvent at room temperature. Chemical shifts are reported in ppm relative to the residual solvent peak. Samples for analysis were prepared as follows: 10 mg of dry analyzed sample was placed in a 5 mm NMR tube, followed by adding MeOH-d4 (0.7 mL). The tube was closed by a stopcock and shacked intensely until a homogeneous solution was obtained.

## 4. Conclusions

TLC-screening combined with HPLC-DAD and NMR proved the reliable and comprehensive method for evaluating the identity, pharmacological value, and composition of the arctic root chosen products. The results indicated the absence of marker compounds in three samples confirmed by TLC, HPLC-DAD, and NMR methods. In four samples presence of rosavin was not detected, which may indicate falsification or enzymatic degradation of the marker. Besides Pharmacopeia standard (S5), only two samples contain all important arctic root markers, which proves their good quality and authenticity. The results indicated the importance of quality control and strict regulations for herbal supplements and preparations.

## Figures and Tables

**Figure 1 molecules-27-08299-f001:**
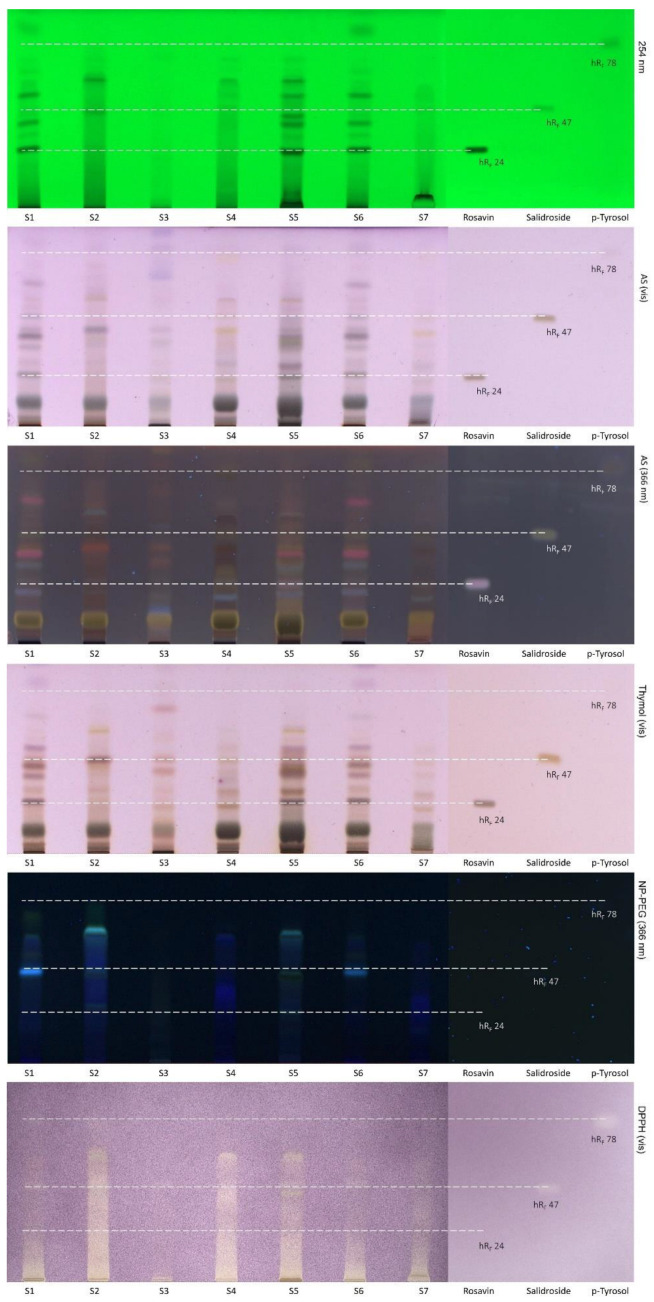
TLC-chemical screening of the arctic root extracts and standards. MP: ethyl acetate-methanol-water (77:13:10, *V*/*V*/*V*).

**Figure 2 molecules-27-08299-f002:**
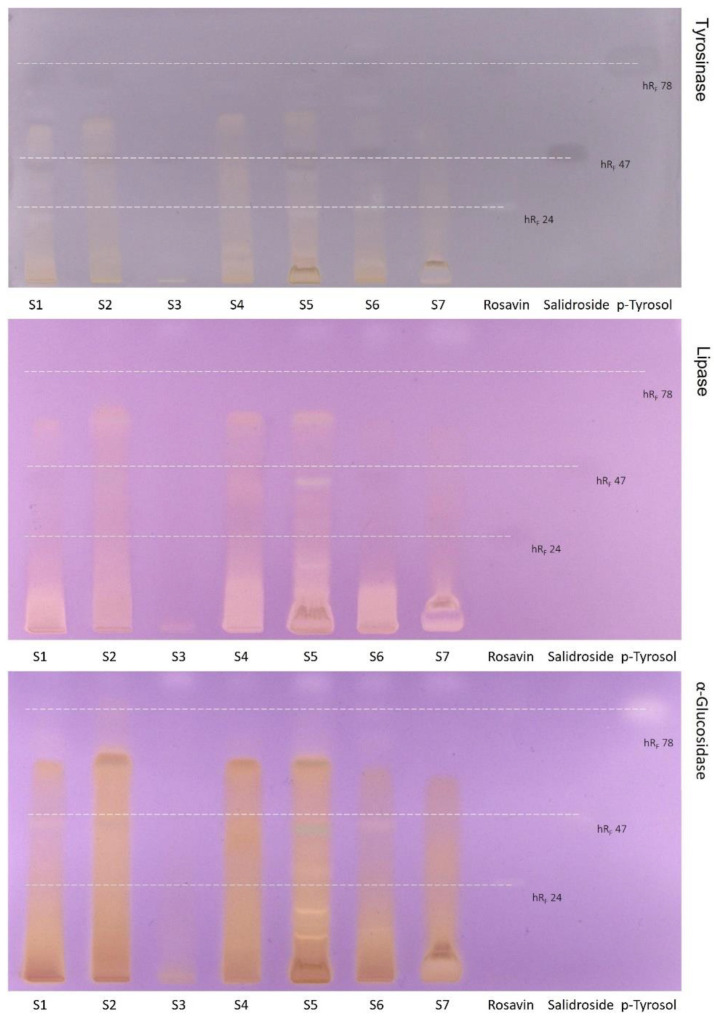
TLC-biochemical screening of the arctic root extracts and standards. MP: ethyl acetate-methanol-water (77:13:10, *V/V/V*).

**Figure 3 molecules-27-08299-f003:**
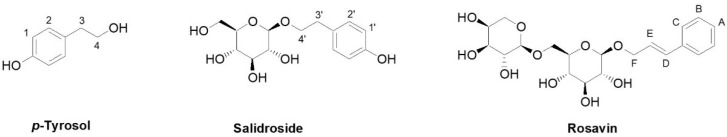
Structures of standard samples p-tyrosol, salidroside, and rosavin) with markings of characteristic signals.

**Figure 4 molecules-27-08299-f004:**
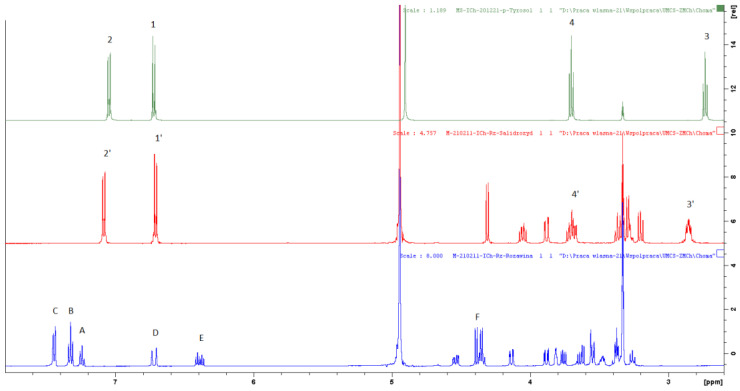
^1^H NMR spectra of standard samples of rosavin (blue), salidroside (red), and p-tyrosol (green) (from the bottom to the top) with markings of characteristic signals.

**Figure 5 molecules-27-08299-f005:**
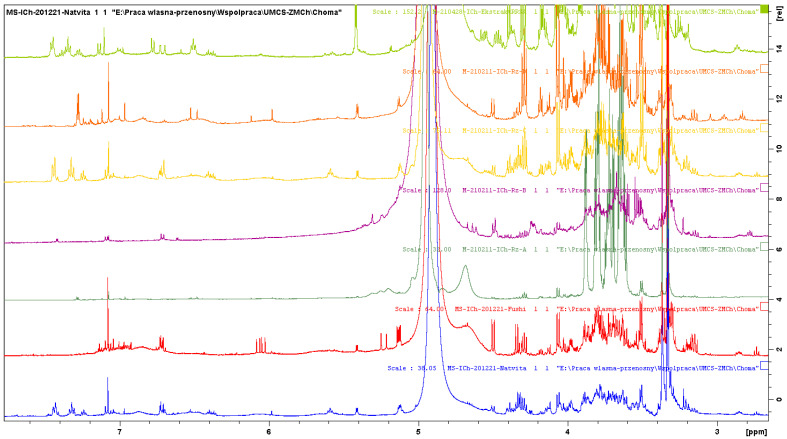
^1^H NMR spectra of S1 (blue), S2 (red), S7 (green), S3 (purple), S6 (yellow), S4 (orange), and S5 (light green) (from the bottom to the top).

**Figure 6 molecules-27-08299-f006:**
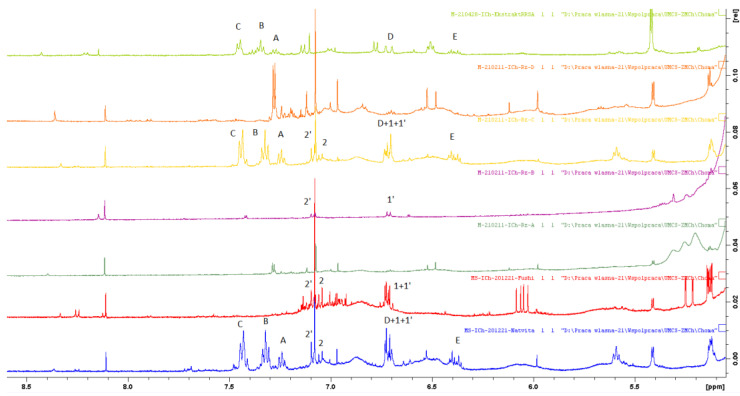
Fingerprint region of ^1^H NMR spectra of S1 (blue), S2 (red), S7 (green), S3 (purple), S6 (yellow), S4 (orange), and S5 (light green) (from the bottom to the top) with markings of characteristic signals.

**Table 1 molecules-27-08299-t001:** HPLC calibration parameters for chosen standards.

Compound	Retention Time (min)	Calibration Range ^2^ (µg/mL)	Regression Equation ^1^	R ^2^ (*n* = 3)	LOD (µg/mL)	LOQ (µg/mL)
**Rosavin**	17.2	1–200	y = 33071x +413,929	R^²^ = 0.993	3.76	11.41
**Salidroside**	4.0	1–200	y = 13110x + 14,399	R^²^ = 0.989	1.04	3.16
**p-Tyrosol**	5.4	1–100	y = 57136x − 186,643	R^²^ = 0.998	0.01	0.02

^1^ y: peak area, x: concentration (µg/mL),^2^ five calibration points.

**Table 2 molecules-27-08299-t002:** The contents of rosavin, salidroside, and p-tyrosol in arctic root extracts.

ID	Content ± SD (µg/mL)		
	Rosavin	Salidroside	p-Tyrosol
**S1**	59.99 ± 3.74	26.37 ± 0.72	2.73 ± 0.50
**S2**	ND	13.12 ± 0.89	4.25 ± 0.33
**S3**	ND	ND	ND
**S4**	ND	ND	ND
**S5**	100.46 ± 5.44	37.86 ± 3.94	4.94 ± 0.12
**S6**	84.71 ± 5.54	21.33 ± 4.51	2.61 ± 0.52
**S7**	ND	ND	ND

ND—not detected

**Table 3 molecules-27-08299-t003:** Presence of marker compounds in the arctic root samples analyzed by NMR.

ID	Rosavin	Salidroside	p-Tyrosol
**S1**	+	+	+
**S2**	ND	+	+
**S3**	ND	Traces	ND
**S4**	ND	Traces	ND
**S5**	+	+	+
**S6**	+	+	+
**S7**	ND	Traces	ND

+—present; ND—not detected.

## Data Availability

Data are contained within the article.

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
