# Peer review of "Qualitative and Quantitative Evaluation of Rosavin, Salidroside, and p-Tyrosol in Artic Root Products via TLC-Screening, HPLC-DAD, and NMR Spectroscopy"

_molecules, 2022, doi:10.3390/molecules27238299_

Round 1

Reviewer 1 Report

The present research paper deals with an important problem in a field of quality control of medicinal plants, in particular arctic root using an array of modern physical and chemical methods of analysis. The paper is well-written and comprehensively substantiated, the results are clear and sufficiently described. I believe that after minor revision of a few technical issues, it can be published.

The issues are as follows:

Lines 161, 164: please use superscript for the isotope designation in the “1H NMR”.

Lines 189 to 199: please indicate the purity of the used reagents (98%? 99.0%? 99.99%?)

Line 301: please recite isotopic purity and origin of the deuterated methanol used.

Section 3.7: please also indicate types of NMR experiments used (especially pulse program types).

Author Response

The present research paper deals with an important problem in a field of quality control of medicinal plants, in particular arctic root using an array of modern physical and chemical methods of analysis. The paper is well-written and comprehensively substantiated, the results are clear and sufficiently described. I believe that after minor revision of a few technical issues, it can be published.

The issues are as follows:

Lines 161, 164: please use superscript for the isotope designation in the “1H NMR”.

Lines 189 to 199: please indicate the purity of the used reagents (98%? 99.0%? 99.99%?)

Line 301: please recite isotopic purity and origin of the deuterated methanol used.

Section 3.7: please also indicate types of NMR experiments used (especially pulse program types). 

Reply: Thank you for pointing this out. We have added lacking information in the manuscript.

All changes are marked in yellow.

Reviewer 2 Report

This can be an interesting article combining TLC HPLC, MS and NMR results, however, there are no HPLC chromatograms, no LC-MS results, no MS spectra…

The NMR data should be provided in a Table. 

Figures with NMR spectra are not described – the signals  are not  depicted by atom numbers etc.

Fig. 4 – the right side of the sepectra could be presented as   minimized – it is not very clear… nor described

Section 3.6. HPLC-DAD   -  The gradient fo HPLC can be written shortly

Section 3.7. NMR – please write wjat kind of spectra were obtained

Please, use lines between sections and section titles.

Author Response

This can be an interesting article combining TLC HPLC, MS and NMR results, however, there are no HPLC chromatograms, no LC-MS results, no MS spectra…

Reply: We have added the chromatogram to the supplement material.

The NMR data should be provided in a Table. 

Reply. Thank you for your suggestion, we added signal description directly on 1H NMR spectra.

Figures with NMR spectra are not described – the signals are not depicted by atom numbers etc.

Reply: Thank you for pointing this out. We have added lacking information in the manuscript. Namely, we added the compounds structure with atoms and signals assigned to it (Fig.3, Fig.4, Fig.6.)

Fig. 4 – the right side of the spectra could be presented as   minimized – it is not very clear… nor described

Reply. Thank you for pointing this out. We added a new Figure 6 with fingerprint region of1H NMR spectra and added signal description.  

Section 3.6. HPLC-DAD -The gradient for HPLC can be written shortly.

Reply. Thank you for pointing this out. We described it shortly.

Section 3.7. NMR – please write what kind of spectra were obtained.

Reply: Thank you for pointing this out. We have added lacking information in the manuscript

Please, use lines between sections and section titles.

Reply. We have added.

All changes are marked in yellow.

Round 2

Reviewer 2 Report

Dear Authors, 

In Table 1 - For Retention time it is enough to write just 1 digit after dot.

In Table 2, please do not use too many significant digits for the data  because their SD are too high...

It will be sufficient to use only 3 digits.

Author Response

In Table 1 - For Retention time it is enough to write just 1 digit after dot.

In Table 2, please do not use too many significant digits for the data  because their SD are too high...

It will be sufficient to use only 3 digits.

Reply: Thank you for your remarks. We have made the appropriate changes.